# Rapid Production and Genetic Stability of Human Mesenchymal Progenitor Cells Derived from Human Somatic Cell Nuclear Transfer-Derived Pluripotent Stem Cells

**DOI:** 10.3390/ijms22179238

**Published:** 2021-08-26

**Authors:** Soo Kyung Jung, Jeoung Eun Lee, Chang Woo Lee, Sung Han Shim, Dong Ryul Lee

**Affiliations:** 1CHA Advanced Research Institute, CHA University Bundang Medical Center, Seongnam 13488, Gyeonggi, Korea; bellotsk@hanmail.net (S.K.J.); jel43@chamc.co.kr (J.E.L.); lcwckddn@chamc.co.kr (C.W.L.); 2Department of Biomedical Science, CHA University, Seongnam 13488, Gyeonggi, Korea; shshim@cha.ac.kr

**Keywords:** human mesenchymal progenitor cells, human SCNT-PSCs, genetic stability, differentiation efficiency

## Abstract

Pluripotent stem cell-derived mesenchymal progenitor cells (PSC-MPCs) are primarily derived through two main methods: three-dimensional (3D) embryoid body-platform (EB formation) and the 2D direct differentiation method. We recently established somatic cell nuclear transfer (SCNT)-PSC lines and showed their stemness. In the present study, we produced SCNT-PSC-MPCs using a novel direct differentiation method, and the characteristics, gene expression, and genetic stability of these MPCs were compared with those derived through EB formation. The recovery and purification of SCNT-PSC-Direct-MPCs were significantly accelerated compared to those of the SCNT-PSC-EB-MPCs, but both types of MPCs expressed typical surface markers and exhibited similar proliferation and differentiation potentials. Additionally, the analysis of gene expression patterns using microarrays showed very similar patterns. Moreover, array CGH analysis showed that both SCNT-PSC-Direct-MPCs and SCNT-PSC-EB-MPCs exhibited no significant differences in copy number variation (CNV) or single-nucleotide polymorphism (SNP) frequency. These results indicate that SCNT-PSC-Direct-MPCs exhibited high genetic stability even after rapid differentiation into MPCs, and the rate at which directly derived MPCs reached a sufficient number was higher than that of MPCs derived through the EB method. Therefore, we suggest that the direct method of differentiating MPCs from SCNT-PSCs can improve the efficacy of SCNT-PSCs applied to allogeneic transplantation.

## 1. Introduction

Mesenchymal stem cells (MSCs) and mesenchymal progenitor cells (MPCs) obtained from fetal and adult tissues have been considered sources in clinical cell therapy for a variety of diseases [1,2]. According to the International Society for Cell Therapy (ISCT), MPCs are plastic-adherent under standard culture conditions and present specific surface antigens such as CD105, CD73, and CD90, but do not express hematopoietic markers such as CD45, CD34, or human leukocyte antigen-DR (HLA-DR). In addition, MPCs must have the potential to differentiate into adipocytes, osteoblasts, and chondrocytes [3]. In fact, MPCs are able to secrete multiple cytokines, growth factors, and exosomes containing microRNAs and other molecules, which affect immune modulation, angiogenesis, apoptosis, cell survival and proliferation [1,2,4]. Therefore, MPC functional mechanisms are most commonly considered to be directly involved in host tissues and paracrine signaling [4]. 

Tissue-derived MPCs have tremendous potential to treat incurable diseases, but there are obstacles to their clinical application: their limited proliferative capacity and different biological characteristics of cells originating from different donors [5,6,7,8]. In addition, many reports have suggested that MPCs with low levels of HLA-DR expression and high levels of immune-modulating activity can be allogeneic grafts that cause no immune rejection [3,9]. However, after transplantation, immune rejection of MPCs or their derivatives recurs, and their therapeutic effect might be lost as their immunomodulatory activity is diminished upon differentiation into functional cells in the host tissues [10]. 

Pluripotent stem cells (PSCs) have the potential to self-renew, proliferate without limit in vitro, and differentiate and produce all kinds of functional cells [11,12,13]. Embryonic stem cells (ESCs) are typical PSCs derived from the inner cell mass of blastocysts and are useful cell sources for research and clinical purposes [14]. However, researchers face a high hurdle in overcoming the immune reaction induced when they use ESCs for clinical application, except in neurological or eye diseases [15], which may contribute to the establishment of dedifferentiated PSCs, such as induced pluripotent stem cells (iPSCs) and somatic cell nuclear transfer-derived PSCs (SCNT-PSCs) through transcription factor-mediated and oocyte-mediated reprogramming [11,12,13]. These two types of PSCs not only have morphology and characteristics similar to those of ESCs but also have matched HLA types when they are transferred into host tissues with the same source of somatic cells. We recently established several SCNT-PSC lines from healthy donors and patients and showed their stemness to produce functionally differentiated cells [10,13,16]. Additionally, it was reported that MPCs and osteoblasts differentiated from HLA-matched homozygous SCNT-PSCs can reduce the immune reaction compared to that induced by HLA-mismatched cells transplanted into humanized mice [10], which may suggest that HLA-matched homozygous SCNT-PSC-MPCs will be useful sources for use in allogeneic cell therapy. 

The functional potentials of all PSC-MPCs are comparable to those of tissue-derived MPCs, but the proliferative potential of PSC-MPCs in vitro is very high and can be produced in sufficient number by mass culturing for use in cell therapy [17]. Various methods of producing PSC-MPCs have been reported, and they can be classified into two main methods: the 3D-platform method (via embryoid body (EB) formation; EB-MPCs) and the 2D method (bypassing EB formation; Direct-MPCs) established before treatment with various growth factors or small molecules to induce differentiation [18,19,20,21]. Generally, compared with the 2D method for producing MPCs directly, the EB-MPC method requires multiple steps and a long culturing time to obtain differentiated and purified MPCs. It has also been reported that homologous recombination (HR), which ensures accurate DNA replication and strand-break repair, is necessary for the survival and maintenance of ESCs. This finding indicates that genomic stability should be sustained through faithful replication and repair of DNA to attain pluripotency. Additionally, HR may contribute to genomic stability during mouse ESC differentiation. In fact, DNA breaks are repaired by homologous recombination (HR)-mediated proteins, such as Rad51 homologous 1 (Rad51) and Rad52, and this repair diminishes the cell death rate [22,23]. Therefore, to extend the efficacy of SCNT-PSC-MPCs in cell therapy, we aimed to enhance an efficient medium using a Rad51 activator and the direct differentiation method, bypassing EB formation, and we analyzed not only their proliferative and differential potential but also their genetic stability during differentiation and cultivation in vitro. 

## 2. Results

### 2.1. Differentiation of SCNT-PSCs into MPCs by the 2D-Direct Method

Two human SCNT-PSC lines (CHA-SCNT-hPSC-17 and hPSC-18) and a conventional human ESC line (CHA-hESC-15) were differentiated into MPCs using the 3D method (CHA-SCNT-hPSC-17-EB-MPCs, CHA-SCNT-hPSC-18-EB-MPCs, and CHA-hESC-15-EB-MPCs) and 2D methods (CHA-SCNT-hPSC-17-Direct-MPCs, CHA-SCNT-hPSC-18-Direct-MPCs, and CHA-hESC-15-Direct-MPC). As shown in our previous reports [21], derivation of EB-MPCs from PSCs required almost 60 days (14 days of EB culture, 16 days of attachment and differentiation, and 25–30 days with as many as 5 passages to obtain a homogeneous population, Figure 1A). To develop a faster and more efficient method of PSC-MPC production, we applied a Direct-MPC induction method using a differentiation medium containing SB431542 (a TGF-β inhibitor), a ROCK inhibitor, and RS-1. Derivation of Direct-MPCs from human PSCs required approximately 34 days (4 days of differentiation and 25–30 days with as many as 5 passages to obtain a homogeneous population, Figure 1A), and this Direct method produced MPCs faster than the EB method. 

All human PSC-MPCs derived from the three PSC lines exhibited similar fibroblastic morphology (Figure 1B), and the conventional karyotypes of all differentiated MPCs were normal (data not shown). The proliferation of all human PSC-MPCs was very similar regardless of the two differentiation methods (Figure 2). Additionally, in all PSC-MPCs, the immunophenotypic markers of pluripotency (TRA-1-60 and SSEA4) and hematopoiesis (CD34 and CD45) were negatively expressed, and those of MPCs (CD29, CD44, CD90, and CD105) were highly expressed (Figure 2). 

We compared the morphology and characteristics of the Direct-MPCs from PSCs according to the number of passages in vitro. The morphology of the CHA-SCNT-hPSC-17-Direct-MPCs and CHA-SCNT-hPSC-18-Direct-MPCs was not changed in early passages (3–5), middle passages (8–10), and late passages (13–14) and was similar to that of the CHA-hESC-15-Direct-MPCs. The immunophenotypic markers of these MPCs showed similar patterns at different passages. Unexpectedly, in the late passage of CHA-SCNT-hPSC-17-Direct-MPC, the expression of CD90 was decreased but was not changed in CHA-SCNT-hPSC-18-Direct-MPCs or CHA-hESC-15-Direct-MPCs. Therefore, we suggested that the characteristics of Direct-MPCs from PSCs were sustained at the late stage but their characteristics varied according to the PSC line (Appendix A).

### 2.2. Differentiation Potential and Functional Analysis of the SCNT-PSC-MPCs

To confirm their differentiation potential, all PSC-MPCs derived through the two differentiation methods were differentiated into mesodermal lineages in vitro. SCNT-PSC-MPCs were effectively differentiated into adipocytes, osteoblasts, and chondrocytes, and their ability to differentiate was very similar to that of ESC-MPCs, regardless of the differentiation method or the source of the PSCs (Appendix A). 

In addition, to confirm the in vivo wound healing potential of SCNT-hPSC-MPCs, we excised skin to create wounds in the back of an STZ-induced severe diabetes mouse model. CHA-SCNT-hPSC-18-Direct-MPCs (1 × 10^6^ cells per mouse) were injected into the wounded mice via the tail vein, and PBS was also injected as a negative control. The wound area was examined around the wound margin and assessed using NIH Image J software. Interestingly, we observed a higher survival rate of severely diabetic mice in the Direct-MPC-injected group than in the PBS-injected control group (66.7% vs. 50% survival at day 18, Figure 3A). Additionally, at day 18, the MPC-injected mice displayed accelerated wound healing compared with PBS-injected mice (13.1 ± 3.1% vs. 32.7 ± 4.6% of remaining wounds; *p* < 0.05, Figure 3B,C). 

### 2.3. Array CGH of SCNT-PSCs-MPCs

Three isogenic MPC sets derived from SCNT-PSC and ESC lines have been established by different methods (Direct and EB methods). To verify the variation in genetic stability during differentiation, CNVs and SNPs in all genome variations of PSCs and its MPCs were analyzed by array CGH. Individual representations of genomic variation in PSCs and their MPC lines are shown in Appendix A. We observed that the genomic variations were mainly located in subtelomeric and pericentromeric regions. 

In our preliminary study, we examined the effect of RS-1, a Rad51 activator, on genome variation during direct differentiation (treatment for 24 h during differentiation on day three) into PSC-MPCs. The concentration was chosen on the basis of a preliminary toxicity test (data not shown) and our previous report [23]. As shown in Appendix A, the addition of 10 μM RS-1 reduced the average number of de novo CNV variations (at >100 kbp resolution) at passage 12 (*p* < 0.05). However, we did not find any difference in >50 kbp resolution; we added RS-1 to the direct differentiation medium for the subsequent study. 

Detailed characteristics of total genomic variations in EB-MPCs and Direct-MPCs are shown for comparison in Figure 4. At passage 5, the average numbers of gain and loss CNVs (at >100 kbp resolution) in EB-MPCs were not different compared to those of Direct-MPCs (5.67 ± 2.08 and 0.33 ± 0.58 vs. 6.33 ± 3.51 and 0.33 ± 0.58; *p* > 0.05, Figure 4A). Additionally, at passage 12, the gain and loss CNVs were very similar to those in early passages and not different in the isogenic MPC sets (5.00 ± 1.00 and 0.33 ± 0.58 vs. 4.67 ± 2.03 and 1.00 ± 1.00; *p* > 0.05, Figure 4A). Moreover, the numbers of de novo CNVs at a higher resolution (>50 kbp) were also not different among SCNT-PSC-MPCs or between SCNT-PSC-MPCs and ESC-MPCs (Figure 4 and Appendix A). Appendix A shows seven recurrent DNA variations observed during MPC differentiation from SCNT-PSC lines. These genomic variations are described according to chromosomal position, size, and genes in the region. 

In addition, we analyzed differentiation method-induced SNP variation between PSCs and isogenic MPCs. As shown in Table 1, Table 2 and Table 3, only 1–2% of the SNP variations in all MPCs were detected during differentiation, and these variations did not differ by differentiation methods or time in culture in vitro until passage 12. These results indicate that there were no differences in the characteristics or proliferation capacity of SCNT-PSC-MPCs and ESC-MPCs on the basis of their derivation by the EB or Direct method.

### 2.4. Analysis of Gene Expression Patterns Using a Microarray of SCNT-PSC-MPCs

To compare the similarity of gene expression among isogenic MPCs according to differentiation method, RNAs were isolated from Direct-MPCs and EB-MPCs at passages 5–7 (CHA-SCNT-hPSC-17-Direct-MPCs vs. CHA-SCNT-hPSC-17-EB-MPCs and CHA-SCNT-hPSC-18-Direct-MPCs vs. CHA-SCNT-hPSC-18-EB-MPCs), and their transcriptomes were analyzed via microarray (Affymetrix Human Gene 2.0 ST Array providing 30,000 genes). Although Direct-MPCs were established faster than EB-MPCs, the gene expression patterns were very similar, as shown in scatter plots (correlation coefficient, R = 0.99 for both CHA-SCNT-Direct-MPCs and CHA-SCNT-EB-MPCs). Moreover, a Gene Ontology (GO) analysis showed the same gene expression pattern when genes were classified by cellular function (Figure 5). 

### 2.5. The Teratoma Assay of SCNT-PSCs-MPCs

CHA-SCNT-hPSC-18-Direct-MPCs and CHA-SCNT-hPSC-18-EB-MPCs were transplanted into the left testis of immunodeficient mice to test the probability of teratoma formation due to remaining undifferentiated PSCs after MPC differentiation. No teratoma formation was observed in any mice 15 weeks after injection with MPCs derived from either method (Appendix A). However, teratomas or teratomas with cysts were clearly formed when undifferentiated CHA-SCNT-hPSC-18 cells were injected (data not shown). Considering the presented data, we suggest that these two protocols for MPC-differentiation from SCNT-hPSCs can generate clinically applicable MPCs.

## 3. Discussions

MPCs are being considered as a clinically useful source of cells for cell therapy in regenerative medicine, and more than 1000 clinical trials are currently registered and performed worldwide (according to clinicaltrials.gov, access date: 1 November 2019). It has been suggested that several features of MPCs, such as high therapeutic effects and capacity for large-scale manufacturing and standardization, meet the requirements for extensive use in clinical application [5,6,7,24]. However, MPCs from diverse tissues do not always fulfil these requirements, and alternative sources are needed. Recently, ESC-MPCs have been used as alternative sources of tissue-derived MPCs due to their high proliferative potential and ease of standardization [17,21]. In the present study, we developed a direct differentiation method for MPCs derived from SCNT-PSCs and observed that their characteristics, such as morphology, expression level of cellular surface markers, and gene expression patterns, are very similar to those of ESC-MPCs derived through conventional differentiation methods via EB formation. In addition, when applying our Direct-MPC induction method, a sufficient amount of functional MPCs to be used in cell therapy can be produced early because the derivation and purification of the PSC-MPCs are accelerated. Moreover, SCNT-hPSC-Direct-MPCs exhibit no differences in genomic variations during differentiation and propagation compared to SCNT-hPSC-EB-MPCs.

When our novel direct differentiation method was applied to generate SCNT-PSC-MPCs, the pure MPCs were obtained approximately four weeks earlier than the pure MPCs obtained through the existing conventional method via EB formation (Figure 1) [21]. To analyze the ability of SCNT-hPSC-Direct-MPCs, we performed a comparison study of various MPCs derived through different differentiation methods and from different PSC lines. Proliferative activity and CD markers expression after purification were not different in the MPCs derived through different differentiation methods or from different PSCs (Figure 1B and Figure 2). To analyze the functional potential of SCNT-hPSC-Direct-MPCs, we confirmed their differentiation into adipocytes, osteoblasts, and chondrocytes in vitro and found no difference by differentiation method (Appendix A). In addition, the injection of SCNT-hPSC-Direct-MPCs increased the survival of severely diabetic mice and accelerated wound healing (Figure 3). SCNT-hPSCs have functional potential similar to that of ESCs and can match HLA types when they are transferred into host tissues with somatic cells from the same source. Interestingly, SCNT-hPSC-18 has a homozygous HLA type, and its derivatives are more useful for allogeneic application because they easily find matching HLA types [10]. In the present study, we confirmed that SCNT-hPSC-18-Direct-MPCs have functional potential as stem cells and show the ability to produce differentiated cells, and the direct differentiation of MPCs combining SCNT-PSCs with homozygous HLA type MPCs may contribute to the wide use of cell therapy in allogeneic transplantation. 

Genomic stability during the maintenance and differentiation of stem cells is very important to their clinical use. In our SCNT-hPSC-Direct-MPCs, we observed normal karyotypes (data not shown) and no difference in de novo CNVs (Figure 4 and Appendix A) or SNP variations (Table 1, Table 2 and Table 3) compared to the original SCNT-PSCs and SCNT-hPSC-EB-MPCs, even though quick and simple direct induction using a TGF inhibitor was applied. Appendix A and Appendix A summarize seven recurrent DNA variations that occurred during MPC differentiation from SCNT-PSC lines when different differentiation methods were applied. Five genomic variations were observed in MPCs derived from CHA-SCNT-PSC-17 cells and two genomic variations were observed in MPCs derived from CHA-SCNT-PSC-18 cells. No genomic variations were observed in MPCs derived from CHA-ESC-15 cells. These results may suggest that genetic variation is due to differences in the PSC lines, not the method of differentiation.

A previous report suggested that DNA breaks can occur during stem cell differentiation and that Rad51-mediated HR may contribute to their repair to reduce the cell death rate [22]. In view of this supposition, we added RS-1, a Rad51 activator, on day three of differentiation and obtained early quantities of differentiated MPCs that could be passaged. In the preliminary study, the addition of RS-1 reduced de novo CNVs when a direct differentiation protocol was applied (Appendix A). Therefore, we suggest that elevated Rad51 activity during the early stage of differentiation contributes to diminished genetic instability and results in high survival of differentiated MPCs. 

The transcriptional profiling of isogenic SCNT-PSC-MPCs obtained through the direct differentiation or EB method showed that the cells had very similar properties (Figure 5A). In fact, the pie chart shows the percentage of genes with significantly different expression among genes related to each gene category in SCNT-hPSC-EB-MPCs and SCNT-hPSC-Direct-MPCs, revealing that there is only a small difference in gene expression between the categories of cell lines (Figure 5B). In addition to the analysis data shown in Figure 2 and Appendix A, the characteristics of isogenic PSC-MPCs appear to be unchanged by differentiation methods, while differences in the used PSCs are apparent.

## 4. Materials and Methods 

### 4.1. Ethics Approval 

All experiments using human PSCs were performed under authorization from the Institutional Review Board for Human Research at the CHA University (1044308-201511-SR-024-06, 1044308-201712-LR-051-03), Seongnam, Korea. In addition, the experimental protocols for the use of animals were approved by the Institutional Animal Care and Use Committee of CHA University (IACUC-190043 (approval date : 1 December 2018), 200087 (approval date : 21 May 2020), 200143 (approval date : 22 June 2020)). 

### 4.2. Human PSCs and Culture

Human SCNT pluripotent cell lines, CHA-SCNT-hPSC-17 [10] and hPSC-18 ([10], Korea Stem Cell Registry code hES12019001), and embryonic stem cell line, CHA-hESC-15 ([25], Korea Stem Cell Registry code hES12010028), were cultured on mitotically inactivated mouse embryonic fibroblasts (MEFs) in DMEM/F12 medium supplemented with 20% KnockOut Serum Replacement (KO-SR), 1% non-essential amino acids (NEAA), 0.1 mM β-mercaptoethanol (β-ME), and 4 ng/mL hrbFGF (ES culture media; all of them from Invitrogen). Human PSCs were mechanically passaged every 5 days under a stereomicroscope (SMZ 645, Nikon, Tokyo, Japan).

### 4.3. Differentiation of Human SCNT-PSCs into MPCs 

Human PSC lines (CHA-SCNT-hPSC-17, CHA-SCNT-hPSC-18, and CHA-hESC-15) were differentiated into MPCs with the 3D method via embryoid body formation (EB protocol) or 2D method via adherent culture (Direct protocol) (Figure 1A). 

For the Direct-MPC differentiation protocol, hPSC were cultured in feeder-free condition with mTeSR^TM^1 (85850, STEMCELL Technology, Vancouver, Canada) and CELLstart^TM^ (A1014201, Invitrogen, Carlsbad, CA, USA). Human PSCs were treated with 1 µM SB431542 (TGF-β inhibitors; S4317, Sigma, St. Louis, MO, USA) in DMEM/F12 supplemented with 20% KO-SR, 1% NEAA, and 0.1 mM β-ME for 3 days. Then, the cells were additionally treated with 1 µM SB431542, 10 µM Y27632 (rho-associated protein kinase (ROCK) inhibitor; SCM075, Millipore, Burlington, MA, USA), and 10 µM RAD51-stimulatory compound 1 (RS-1, Rad51 activator; R9782, Sigma, St. Louis, MO, USA) simultaneously. After 24 h, the cells were passaged using 0.05% trypsin-EDTA (25300-054, Invitrogen) onto new CELLstart^TM^ coating culture dish containing MPC media (DMEM/F12 supplemented with 10% fetal bovine serum (FBS; 16000-044, Invitrogen), 1% NEAA, 1% penicillin/streptomycin (P/S; 15140-122, Invitrogen), and 0.1 mM β-ME) (passage 0). We got a homogenous cell population by serial sub-passaging using trypsin-EDTA.

Human PSC-EB-MPCs were produced with a previously described method [21]. In brief, hPSCs were manually harvested as small clumps by using a sterile micro tip and then placed into a low attachment 6-well plate containing EB media (ES culture media without hrbFGF) for suspension culture. From the next day, 1 µM SB431542 was treated for 14 days, and then the formed EBs were plated onto a 0.1% gelatin (G1393, Sigma)-coated 6-well culture dish (140675, Nunc, Waltham, MA, USA) in low glucose media (DMEM low glucose (11885-084, Invitrogen) supplemented with 10% FBS and 1% P/S). After 16 days, the outgrowth MPCs obtained from the EBs were detached by treating 0.05% trypsin-EDTA and transferred into 0.1% gelatin-coated 75 cm^2^ culture flask (156499, Nunc) containing MPC media (passage 0). We got a homogenous cell population by serial sub-passaging using trypsin-EDTA.

The hPSC-MPCs were passaged using trypsin-EDTA when they reached 80–90% confluency. The hPSC-MPCs were checked for their MPC characteristics and cryopreserved in freezing media (10% DMSO (D2650, Sigma), 30% FBS, 60% MPC media) around passage 5.

### 4.4. Growth Kinetics of Human PSC-Derived MPC 

For the analysis of cell growth kinetics, hPSC-derived MPCs were harvested using trypsin-EDTA and seeded onto a 75 cm^2^ culture flask at a density of 400,000 cells. Cell counting by using a hemacytometer was performed at every passage, and the cumulative population doublings between cell passages were evaluated as previously described [26].

### 4.5. Characterization of Human PSC-Derived MPC 

For flow cytometry analysis, hPSC-derived MPCs were fixed for 1 h in pre-chilled 4% paraformaldehyde solution (PFA; BP-031, elBio, Seongnam, Korea) at 4 °C. Then, the cells were washed with fluorescence-activated cell sorting (FACS) buffer composed with 2% FBS in DPBS without Ca^2+^Mg^2+^ (Invitrogen) and incubated with antibodies at room temperature for 30 min in the dark. We used phycoerythrin (PE)-conjugated mouse anti-human TRA-1-60 (560193, BD) and allophycocyanine (APC)-conjugated mouse anti-human/mouse SSEA4 (FAB/435A, R&D systems, Minneapolis, MN, USA) as the stemness markers; APC-conjugated mouse anti-human CD34 (555824, BD) and APC-conjugated mouse anti-human CD45 (555485, BD, Franklin Lakes, NJ, USA) were used as the hematopoietic markers; APC-conjugated mouse anti-human CD29 (559883, BD), APC-conjugated mouse anti-human CD44 (559942, BD), APC-conjugated mouse anti-human CD90 (561971, BD), and APC-conjugated mouse anti-human CD105 (562408, BD) were used as the MPC markers. For negative controls, proper isotype controls and unstained controls were used. After washing, the cells were analyzed using an Accuri C6 Plus flow cytometer equipped with Cell Quest software (BD Biosciences). 

For adipogenic differentiation, confluent hPSC-derived MPCs were cultured in the adipogenic differentiation media (A10070-01, Invitrogen) for 21 days. After adipogenic induction, these differentiated cells were stained with an Oil Red O solution (IW3008, IHC world, Woodstock, MD, USA) according to the manufacturer’s instruction to validate the generation of lipid droplets. For osteogenic differentiation, confluent hPSC-derived MPCs were cultured in the osteogenic differentiation media (A10072-01, Invitrogen) for 21 days. After osteogenic differentiation, these cells were stained with an Alizarin Red solution (IW3001, IHC world) according to the manufacturer’s instructions to examine calcium deposit. For chondrogenic differentiation, hPSC-derived MPCs were seed into a 15 mL conical tube (352096, Falcon, Corning, NY, USA) at a concentration of 5 × 10^5^ cells in a chondrogenesis medium (A10071-01, Invitrogen) for approximately 28 days. Chondrogenic pellets were fixed and then embedded in paraffin. The sectioned samples were stained with an Alcian Blue staining solution (IW3000, IHC World) according to the manufacturer’s instructions to examine the accumulation of glycosaminoglycan.

### 4.6. Teratomas Formation Assay 

Human PSC-derived MPCs, CHA-SCNT-hPSC-18-EB-MPCs, and CHA-SCNT-hPSC-18-Direct-MPCs were assayed for tumorigenic potential by a teratoma assay. Approximately 1 × 10^6^ of cells (at passage 6) were injected into the left testicle of a NOD/SCID male mouse (Labolatory Animal Resource Center, KRIBB; Chungcheongbuk-do, Korea). The right testis had no injection as a normal control. After 15 weeks, the testes were excised, fixed in 4% PFA, embedded in paraffin, sectioned, and then analyzed histologically after hematoxylin-eosin staining. 

### 4.7. Excisional Wound Splinting Model and Cell Transplantation in Type I Diabetes Mouse

Male NOD/SCID mice (12–14 weeks old) were intraperitoneally injected with 40 mg/kg streptozotocin (STZ; S0130, Sigma) dissolved in 0.1 M sterile citrate buffer (pH 4.5). STZ was administered for 3 consecutive days during the first week of the study. Blood was collected from the tail vein for 4–5 weeks after the injection, and blood glucose concentration was measured using a glucose analyzer (ACCU-CHEK Perfoma, Roche, Basel, Switzerland). Mice with blood glucose levels >250 mg/dL were considered diabetic and used for wound experiments.

Male NOD/SCID STZ-induced diabetic mice were randomly divided into 2 groups: PBS-injected (Sham, *n* = 8) and hPSC-Direct-MPC-injected (Direct-MPC, *n* = 9). The excisional wound splinting model was generated as described previously [27,28]. There were no significant differences between the blood glucose levels of the Sham (429.1 ± 35.2) and Direct-MPC (392.1 ± 36.0) (*p* = 0.48). Briefly, all animals were anesthetized by isoflurane (~2%). The hair was removed from the dorsal surface the day before wounding. Two 5 mm thick excision skin wounds were created on each side of the midline using a surgical punch (50.005, Gyneas, Asnières-sur-Seine, France). A splint was placed (476687, Grace Bio-Labs, Bend, OR, USA), instant-bonding adhesive was applied on one side around wound carefully, and then the splint was fixed by sutures. PBS (150 µL) or CHA-SCNT-hPSC-18-Direct-MPCs (1 × 10^6^ cells in 150 µL PBS) were injected into the tail vein. For wound analysis, photographs were obtained at days 1, 3, 7, 11, 14, and 18, and the wound size was measured using Image J. The rate of survival and wound closure was evaluated. The percentage of wound closure was calculated as follows: (original wound area−actual wound area)/(original wound area) × 100.

### 4.8. Array-Based Comparative Genomic Hybridization (Array-CGH) for Genetic Stability

We performed a high-resolution array-CGH to know if there was a difference in genetic stability between hPSC-derived MPCs by differentiation protocol (CHA-hESC-15-EB, Direct-MPC; CHA-SCNT-hPSC-17 EB, Direct-MPC; CHA-SCNT-hPSC-18-EB, Direct-MPC) or between early (p5) and late (p12) passages. 

Copy number variations (CNVs) and single nucleotide polymorphisms (SNP) were analyzed using Affymetrix CytoScan^®^ High-Density Arrays, which include 2.67 million markers for copy number (CN) analysis, including 750,000 biallelic SNP probes and 1.9 million non-polymorphic probes for comprehensive whole-genome coverage. All experimental procedures were performed according to the manufacturer’s instructions (Affymetrix, Santa Clara, CA, USA) by BioCore (Seoul, Korea). The procedure included genomic DNA extraction, digestion and ligation, PCR amplification, PCR product purification, quantification and fragmentation, labeling, array hybridization, washing, and scanning with the GCS 3000 platform (Affymetrix, Santa Clara, CA, USA). All data were visualized and analyzed with the Chromosome Analysis Suite (ChAS) software package (Affymetrix) using Human Genome build hg38. The reporting threshold was set at 100 kb with marker count ≥50.

### 4.9. Micro-Array for Transcriptome Profiling

We performed a micro-array (GeneChip^®^ Human Gene 2.0 ST Arrays, Applied Biosystems, Waltham, MA, USA) to know if there was a difference in gene expression between hPSC-derived MPCs by differentiation protocol (CHA-SCNT-hPSC-17-EB, Direct-MPC; CHA-SCNT-hPSC-18-EB, Direct-MPC) (around passage 7).

Total RNA was isolated using Trizol reagent (Invitrogen). RNA quality was assessed with an Agilent 2100 bioanalyzer (Agilent Technologies, Waltham, MA, USA), and quantity was determined with an ND-1000 spectrophotometer (NanoDrop Technologies, Waltham, MA, USA). RNA samples were used as input into the Affymetrix procedure as recommended by the protocol (http://www.affymetrix.com, access date: 1 November 2019). Briefly, total RNA from each sample was converted to double-strand cDNA. Using a random hexamer incorporating a T7 promoter, amplified RNA (cRNA) was generated from the double-stranded cDNA template though an in vitro transcription (IVT) reaction and purified with the Affymetrix sample cleanup module. cDNA was regenerated through a random-primed reverse transcription using a dNTP mix containing dUTP. The cDNA was then fragmented by UDG and APE 1 restriction endonucleases and end labeled by terminal transferase reaction incorporating a biotinylated dideoxynucleotide. Fragmented end-labeled cDNA was hybridized to the Affymetrix arrays for 16 h at 45 °C and 60 rpm as described in the Gene Chip Whole Transcript (WT) Sense Target Labeling Assay Manual (Affymetrix). After hybridization, the chips were stained using SAPE (Streptavidin Phycoerythrin) and washed in a Genechip Fluidics Station 450 (Affymetrix) and scanned using a Genechip Array scanner 3000 7G (Affymetrix). After the final washing and staining step, an Affymetrix array was scanned using an Affymetrix Model 3000 G7 scanner, and the image data was extracted through Affymetrix Command Console software1.1. The raw .CEL file generated through the abovementioned procedure meant expression intensity data and was used for the next step. Expression data were generated by Transcriptome Analysis Console 4.0.1. For the normalization, a Robust Multi-Average (RMA) algorithm implemented in Transcriptome Analysis Console software was used. Data mining and graphic visualization were performed using ExDEGA (Ebiogen Inc., Seoul, Korea). All microarray data generated in this study were deposited in the Gene Expression Omnibus database (GSE182415).

### 4.10. Statistical Analysis

All data are presented as means ± SE. Statistical analyses were performed using the Student’s *t*-test. The differences were considered significant when *p* < 0.05.

## 5. Conclusions

In conclusion, we found that SCNT-PSC-MPCs can be derived through a simple direct method using differentiation medium with a TGF-β inhibitor and Rad51 activator. Direct-MPCs exhibited similar differentiation potential and proliferative properties, but their derivation was faster and the rate of reaching a sufficient cell count was faster than conventional methods via EB formation. In addition, SCNT-PSC-Direct-MPCs undergoing accelerated differentiation sustained fewer CNVs and SNPs during differentiation and cultivation, which appears to have been due to the addition of RS-1, which enhances genomic stability. On the basis of these results, we suggest that the direct differentiation method of MPCs can contribute to improvements in the efficacy of SCNT-PSCs as therapeutic cells for allogeneic transplantation and establishing a cell bank for suitable application to treatments of incurable diseases. 

## Figures and Tables

**Figure 1 ijms-22-09238-f001:**
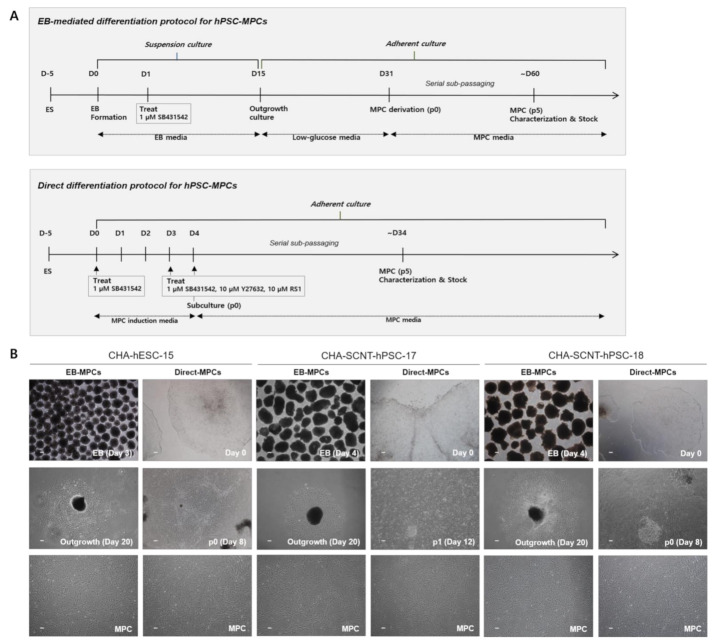
Differentiation procedure and representative morphologies of human pluripotent stem cells differentiated into mesenchymal progenitor cells (hPSC-MPCs). (**A**) Schematic diagram of the differentiation procedures used to generate EB/Direct-MPCs derived from hPSCs. (**B**) Representative phase contrast microscopic images of hPSC-EB-MPCs and hPSC-Direct-MPCs. Bars, 100 μM (magnification, 40×).

**Figure 2 ijms-22-09238-f002:**
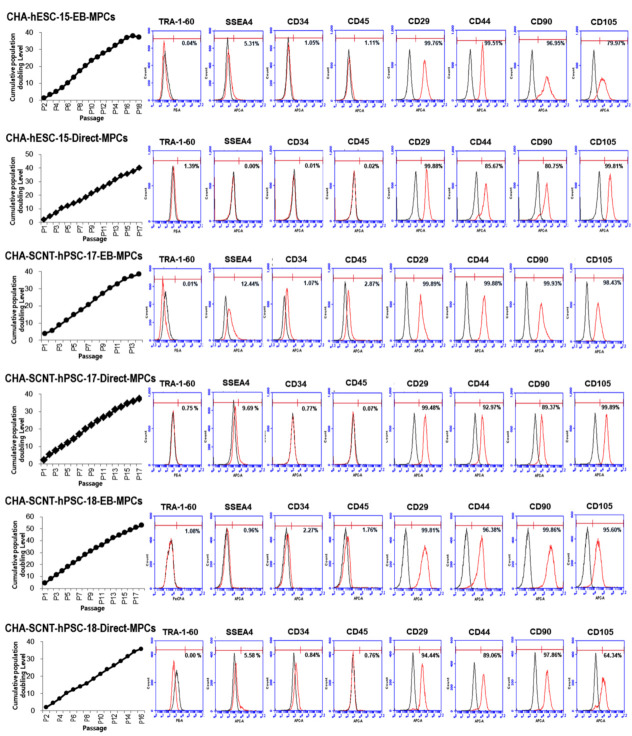
Cumulative population doubling level (CPDL) and expression of surface antigen profiles of hPSC-MPCs. Human PSC-derived MPCs were seeded at 4 × 10^5^ cells per 75 cm^2^ culture flask and counted and passaged when they were 80–90% confluent (approximately 5 days after plating). The proliferative capacities of the hPSC-MPCs are expressed as CPDL on the basis of the formula CPDL = ln(Nf/Ni)ln2, where Ni and Nf are the initial and final cell numbers, respectively, and ln refers to the natural logarithm. Surface antigen expression in hPSC-MPCs (passages 5 and 6) as determined by FACS showed that the markers for MPCs (CD29, CD44, CD90, and CD105) were highly expressed in both EB-MPCs and D-MPCs, but neither pluripotent stem cell markers (TRA-1-60 and SSEA4) nor hematopoietic markers (CD34 and CD45) were highly expressed.

**Figure 3 ijms-22-09238-f003:**
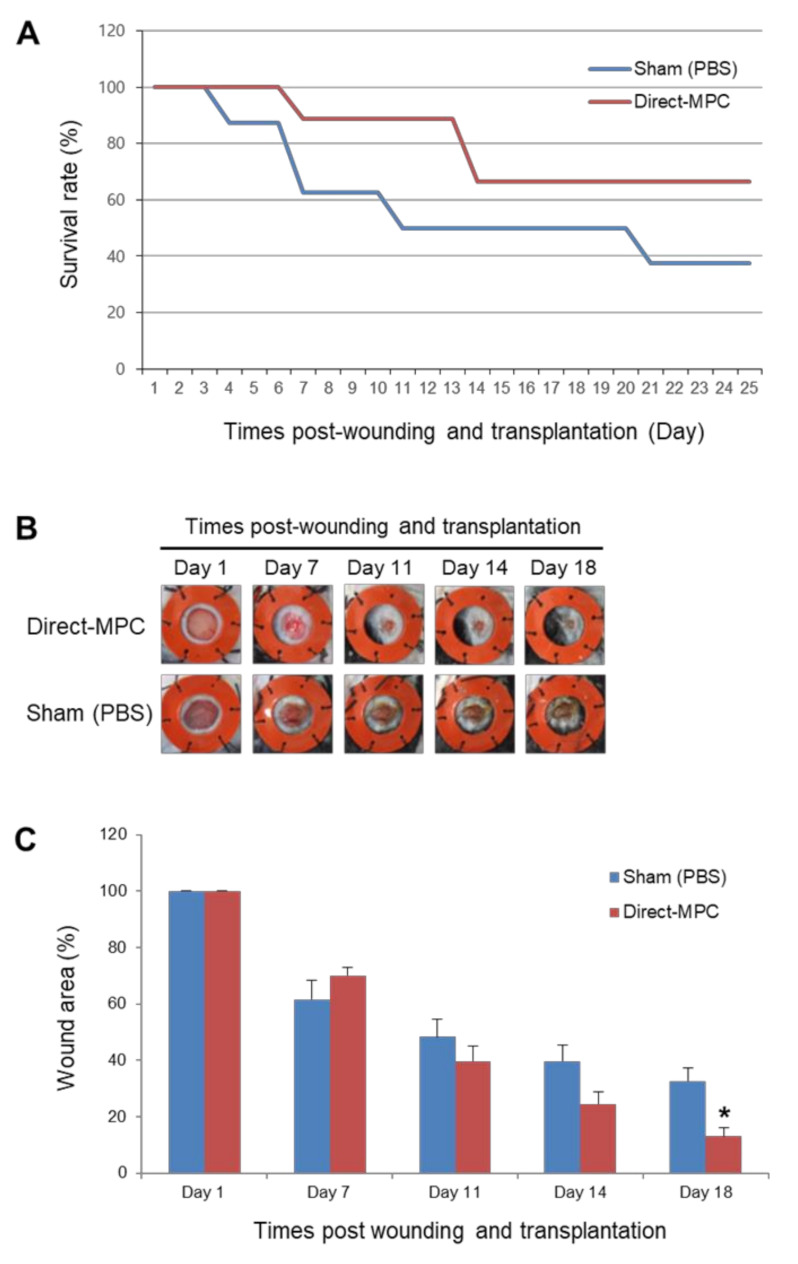
Effects of hPSC-Direct-MPCs on survival and in vivo wound closure in a streptozotocin (STZ)-induced diabetic mouse model. (**A**) Survival rate of both groups of MPCs in STZ-diabetic mice. Sham (PBS) group, *n* = 8; Direct-MPC group, *n* = 9. (**B**) Representative photographs of the wound splinting model established with STZ-diabetic mice after transplantation of PBS (sham control) and CHA-SCNT-hPSC-18-Direct-MPCs on days 1, 7, 11, 14, and 18. (**C**) Wound measurement of the STZ-diabetic mice in each group. Sham (PBS) group, *n* = 4; Direct-MPC group, *n* = 6. * *p* < 0.01.

**Figure 4 ijms-22-09238-f004:**
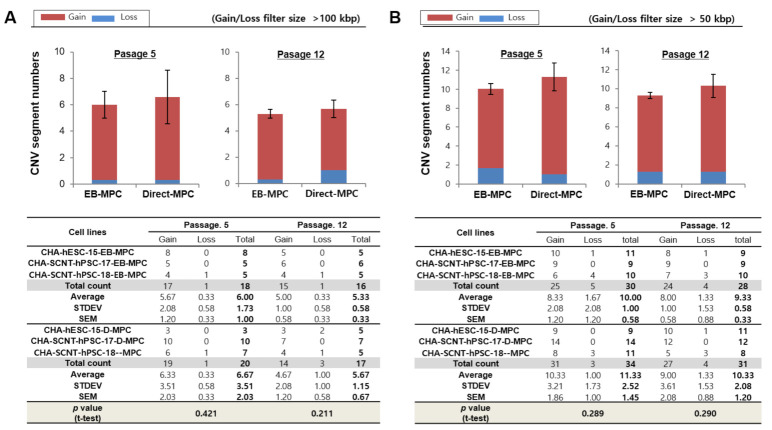
Array-based comparative genomic hybridization analysis of hPSC-derived EB-MPCs and Direct-MPCs at an early passage (5) and late passage (12). Human PSCs were used as the control cells in the genomic variation analysis by array-CGH. The CNV results are presented as the means for MPCs derived from three cell lines: CHA-hESC-15, CHA-SCNT-hPSC-17, or CHA-SCNT-hPSC-18. The data are presented as the means ± SE. (**A**) Variation size >100 kbp, (**B**) variation size >50 kbp.

**Figure 5 ijms-22-09238-f005:**
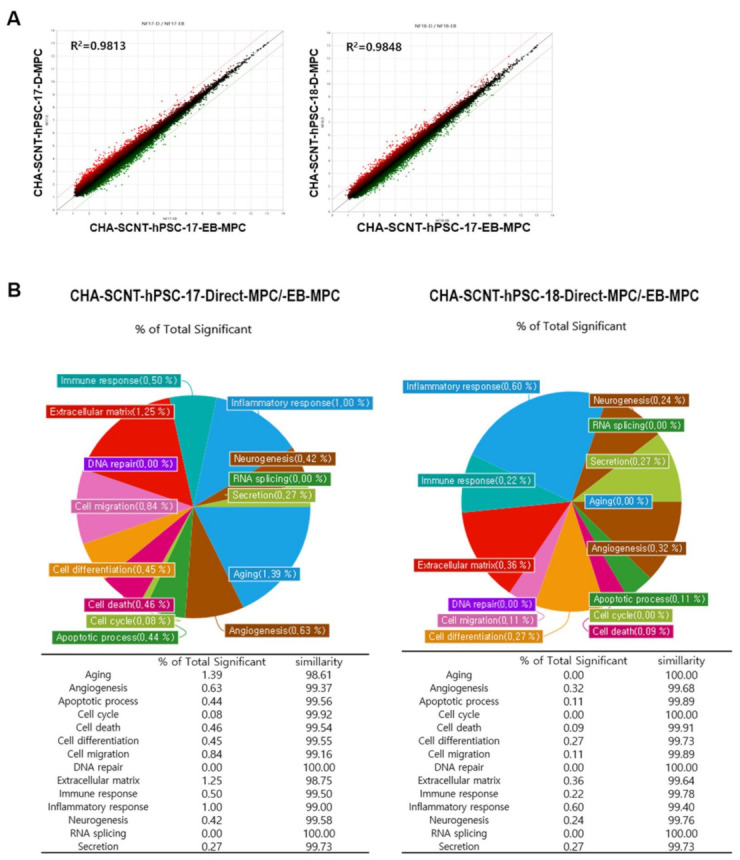
Correlation of gene expression profiles between EB-MPCs and Direct-MPCs derived from the same human SCNT-hPSC line. (**A**) Scatter plot comparing microarray gene expression data of EB-MPCs and Direct-MPCs. The x and y values on the scatter plot are the average normalized signal values shown on a log2 scale. The red and green lines are fold change lines with a default change of 2.0. (**B**) Pie chart showing the percentage of genes with significantly different expression among genes that are related to each gene category in the EB-MPCs and Direct-MPCs.

**Table 1 ijms-22-09238-t001:** SNP analysis of CHA-hESC-15 derived MPCs by array-based comparative genomic hybridization.

CHA-hESC-15	ES (Control)		EB-MPC (p5)	EB-MPC (p12)	Direct-MPC (p5)	Direct-MPC (p12)	
Total SNP	748,953		748,953	748,953	748,953	748,953	
Called SNP	731,057		725,854	729,329	722,029	728,760	
All called SNP	682,461		682,461	682,461	682,461	682,461	
Matched SNP	682,461		678,727	679,629	675,301	678,123	
Different SNP	0		3,734	2,832	7,160	4,338	
Matched %	100%		99.453%	99.585%	98.951%	99.364%	
**CHA-hESC-15**	**Different SNP**	**intron**	**missense**	**coding-synonym**	**Untranslated**	**Un-known**	**other**
EB-MPC (p5)	3734	1437	18	14	18	2158	89
EB-MPC (p12)	2832	1077	11	9	9	1642	85
Direct-MPC (p5)	7160	2844	20	32	48	4023	193
Direct-MPC (p12)	4338	1692	19	13	17	2481	116
Total	11,595	4581	40	46	68	6514	346

**Table 2 ijms-22-09238-t002:** SNP analysis of CHA-SCNT-hRSC-17 derived MPCs by array-based comparative genomic hybridization.

CHA-SCNT-hPSC-17	ES (Control)		EB-MPC (p5)	EB-MPC (p12)	Direct-MPC (p5)	Direct-MPC (p12)	
Total SNP	748,953		748,953	748,953	748,953	748,953	
Called SNP	717,149		721,534	713,570	720,532	716,621	
All called SNP	646,014		646,014	646,014	646,014	646,014	
Matched SNP	646,014		632,815	629,481	632,707	628,918	
Different SNP	0		13,199	16,533	13,307	17,096	
Matched %	100%		97.957%	97.441%	97.940%	99.354%	
**CHA-SCNT** **-hPSC-17**	**Different SNP**	**intron**	**missense**	**coding-synonym**	**Untranslated**	**Un-known**	**other**
EB-MPC (p5)	13,199	4861	35	35	83	7830	355
EB-MPC (p12)	16,533	6204	39	56	91	9720	423
Direct-MPC (p5)	13,307	4899	33	47	76	7927	325
Direct-MPC (p12)	17,096	6467	58	51	106	9982	432
Total	30,607	11,668	88	93	183	17,738	837

**Table 3 ijms-22-09238-t003:** SNP analysis of CHA-SCNT-hRSC-18 derived MPCs by array-based comparative genomic hybridization.

CHA-SCNT-hPSC-18	ES (Control)		EB-MPC (p5)	EB-MPC (p12)	Direct-MPC (p5)	Direct-MPC (p12)	
Total SNP	748,953		748,953	748,953	748,953	748,953	
Called SNP	736,591		728,309	734,471	736,213	729,502	
All called SNP	699,127		699,127	699,127	699,127	699,127	
Matched SNP	699,127		699,163	697,553	697,554	694,450	
Different SNP	0		2964	1574	1573	4667	
Matched %	100%		99.576%	99.775%	99.775%	99.331%	
**CHA-SCNT** **-hPSC-18**	**Different SNP**	**intron**	**missense**	**coding-synonym**	**Untranslated**	**Un-known**	**other**
EB-MPC (p5)	2964	1190	12	12	24	1652	74
EB-MPC (p12)	1574	575	4	5	17	919	54
Direct-MPC (p5)	1573	607	8	4	7	903	44
Direct-MPC (p12)	4677	1850	20	16	29	2616	146
Total	7871	3158	30	27	62	4340	254

## Data Availability

All data generated or analyzed during this study are included in this published article and its Appendix A files. All microarray data generated in this study were deposited in Gene Expression Omnibus database (GSE182415).

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
