# Peer review of "Rapid Production and Genetic Stability of Human Mesenchymal Progenitor Cells Derived from Human Somatic Cell Nuclear Transfer-Derived Pluripotent Stem Cells"

_ijms, 2021, doi:10.3390/ijms22179238_

Round 1
Reviewer 1 Report
The present study evaluates the differentiation potential of stem cells, namely three different cell lines CHA-SCNT-hPSC-17, hPSC-18, CHA-hESC-15. The preliminary stages of differentiation i.e. cell aggregation was performed according to established methods commonly used in embryonic stem cells. The results suggest the potential and functionality of these cells. The research is conducted according to established methods, and for the gene expression pattern the authors again use a well-established method, namely microarray, as in similar studies described in the literature. I believe that the study is complex enough to be published but in the section on evaluating the formation of teratomas, I consider that it would be more suggestive to add histological sections to highlight those described in this study.
Another remark that should be reviewed concerns the use of the term embryoid bodies. There are places where the authors use for EB embryoid body but also in the text for EB appears the expression embryonic body (line 14).
Author Response
Reviewer#1.
The present study evaluates the differentiation potential of stem cells, namely three different cell lines CHA-SCNT-hPSC-17, hPSC-18, CHA-hESC-15. The preliminary stages of differentiation i.e. cell aggregation was performed according to established methods commonly used in embryonic stem cells. The results suggest the potential and functionality of these cells. The research is conducted according to established methods, and for the gene expression pattern the authors again use a well-established method, namely microarray, as in similar studies described in the literature.
=> Thank you for reviewer’s good comment.
I believe that the study is complex enough to be published but in the section on evaluating the formation of teratomas, I consider that it would be more suggestive to add histological sections to highlight those described in this study.
=> In the text regarding teratoma analysis, it seems to be a little vague in description. So we changed the description as below. “However, teratomas or teratomas with cysts were clearly formed in CHA-SCNT-hPSC-18 cells (data not shown).” has been changed to “However, teratomas or teratomas with cysts were clearly formed when undifferentiated CHA-SCNT-hPSC-18 cells were injected (data not shown). (Line 335-336). Also, recently, we have published several papers regarding PSC-MPCs and their characteristics (PMID: 34332643, PMID: 34073789, PMID: 32586410, and PMID: 30896075). In all paper, we confirmed teratoma were not formed when hPSC-MPC were injected into immunodeficient mice.
Another remark that should be reviewed concerns the use of the term embryoid bodies. There are places where the authors use for EB embryoid body but also in the text for EB appears the expression embryonic body (line 14).
=> Thank you for reviewer’s good comment. So we all changed to “embryoid body” or “via EB formation” to clarify the meaning (highlighted line 17, 21, 73, 122, 350, 357, and 404).

Reviewer 2 Report
General Comments: In this study, the authors describe the derivation of mesenchymal progenitor cells (MPCs) from somatic cell nuclear transfer (SCNT)-pluripotent stem cell (PSC) lines using a novel direct differentiation method involving the activation of homologous recombination-mediated proteins like Rad51. In addition, they also present a comparative analysis of MPCs derived via their direct differentiation method alongside the traditional embryoid body (EB) method from both, SCNT-PSC lines as well as ES cell lines. While the authors find no major differences between these methods or cells lines in terms of MPC morphology, proliferation, expression of immunophenotypic markers, trilineage differentiation potential in vitro or in vivo, genomic variations or gene expression patterns, they do find that their novel direct differentiation method markedly accelerates the generation of pure MPCs by nearly 4 weeks over the conventional EB method. This reviewer feels that from a therapeutic standpoint, gaining time in the order a month could prove to be important for the recovery of an individual where these MPCs may be utilized. For instance, there are athletes who need surgeries to repair connective tissue injuries. Some surgeons inject autologous MPCs harvested from the bone marrow, then expanded and purified, to improve surgical outcomes. There are limitations to this method in the number of cells that can be harvested or expanded over time. There is also the issue of MPCs losing their “stemness” over longer culture periods. The technique described by the authors could be useful in such a scenario as it allows for the expansion of the patient-derived iPSC lines to get sufficient starting material followed by direct differentiation to MPCs.
Specific Comments: are as below:
- For Figure 1B, it would help to drive the point home about the accelerated rate of MPC derivation from the direct differentiation method if the authors could add the days of culture for each of the microscope images and not just say Day 0 or p0 p1.
- Figure 1, legend says bars but I did not see any clear scale bars in any of the images. Please fix that.
- For Figure 2, the population doubling charts are depicted well but the flow analysis layout is cumbersome to scroll up and down to compare between lines. The details charts could perhaps be put in the supplement and a more concise version could be developed to showcase the differences and similarities.
- In Figure 2, the authors address the variation in CD90 expression but do not mention anything about the seemingly higher SSEA-4 expression in MPCs derived from the hPSC-17 line. Does that imply that these MPCs may have more pluripotent cells and therefore have a greater teratoma forming capacity? The only line used for the teratoma assay is the hPSC-18 line.
Author Response
Reviewer#2.
General Comments: In this study, the authors describe the derivation of mesenchymal progenitor cells (MPCs) from somatic cell nuclear transfer (SCNT)-pluripotent stem cell (PSC) lines using a novel direct differentiation method involving the activation of homologous recombination-mediated proteins like Rad51. In addition, they also present a comparative analysis of MPCs derived via their direct differentiation method alongside the traditional embryoid body (EB) method from both, SCNT-PSC lines as well as ES cell lines. While the authors find no major differences between these methods or cells lines in terms of MPC morphology, proliferation, expression of immunophenotypic markers, trilineage differentiation potential in vitro or in vivo, genomic variations or gene expression patterns, they do find that their novel direct differentiation method markedly accelerates the generation of pure MPCs by nearly 4 weeks over the conventional EB method. This reviewer feels that from a therapeutic standpoint, gaining time in the order a month could prove to be important for the recovery of an individual where these MPCs may be utilized. For instance, there are athletes who need surgeries to repair connective tissue injuries. Some surgeons inject autologous MPCs harvested from the bone marrow, then expanded and purified, to improve surgical outcomes. There are limitations to this method in the number of cells that can be harvested or expanded over time. There is also the issue of MPCs losing their “stemness” over longer culture periods. The technique described by the authors could be useful in such a scenario as it allows for the expansion of the patient-derived iPSC lines to get sufficient starting material followed by direct differentiation to MPCs.
=> Thank you for reviewer’s good comment.
Specific Comments: are as below:
- For Figure 1B, it would help to drive the point home about the accelerated rate of MPC derivation from the direct differentiation method if the authors could add the days of culture for each of the microscope images and not just say Day 0 or p0 p1.
=> Thank you for reviewer’s good comment. We modified Fig 1B (added the days of culture)
- Figure 1, legend says bars but I did not see any clear scale bars in any of the images. Please fix that.
=> Thank you for reviewer’s good comment. We modified Fig 1B (enlarged the scale bars).
- For Figure 2, the population doubling charts are depicted well but the flow analysis layout is cumbersome to scroll up and down to compare between lines. The details charts could perhaps be put in the supplement and a more concise version could be developed to showcase the differences and similarities.
=> Thank you for reviewer’s comment. However, in this phase, it seems to be not easy work.
- In Figure 2, the authors address the variation in CD90 expression but do not mention anything about the seemingly higher SSEA-4 expression in MPCs derived from the hPSC-17 line. Does that imply that these MPCs may have more pluripotent cells and therefore have a greater teratoma forming capacity? The only line used for the teratoma assay is the hPSC-18 line.
=> Thank you for reviewer’s comment. We have found some difference of CD90 expression among PSC-MPCs. However, we didn’t find any difference their function according to the expression of CD90 and any tendency towards the sources of the cell lines. Thus, the difference in the expression of CD90 is now considered to be a difference in individual characteristics rather than a difference in functionality. So. in the text of the Results, we already described this situation (line 268-272)
